# Using Time Series Optical and SAR Data to Assess the Impact of Historical Wetland Change on Current Wetland in Zhenlai County, Jilin Province, China

**Sixue Shi** [1,2], **Yu Chang** [1,*], **Yuehui Li** [1], **Yuanman Hu** [1], **Miao Liu** [1], **Jun Ma** [3], **Zaiping Xiong** [1], **Ding Wen** [4], **Binglun Li** [1,2] **and Tingshuang Zhang** [1,2]

1   CAS Key Laboratory of Forest Ecology and Management, Institute of Applied Ecology,
    Chinese Academy of Sciences, Shenyang 110016, China; shisixue15@mails.ucas.ac.cn (S.S.);
    liyh@iae.ac.cn (Y.L.); huym@iae.ac.cn (Y.H.); lium@iae.ac.cn (M.L.); zaipingx@iae.ac.cn (Z.X.);
    liblqz@163.com (B.L.); zhangtingshuang15@mails.ucas.ac.cn (T.Z.)
2   College of Resources and Environment, University of Chinese Academy of Sciences, Beijing 100049, China
3   Ministry of Education Key Laboratory for Biodiversity Science and Ecological Engineering, Coastal
    Ecosystems Research Station of the Yangtze River Estuary, and Shanghai Institute of EcoChongming (SIEC),
    Fudan University, Shanghai 200433, China; ma_jun@fudan.edu.cn
4   South China Institute of Environmental Science, Ministry of Ecology and Environment,
    Guangzhou 510655, China; wending@scies.org
*   Correspondence: changyu@iae.ac.cn; Tel.: +86-024-8397-0351

**Abstract:** Wetlands, as the most essential ecosystem, are degraded throughout the world. Wetlands in Zhenlai county, with the Momoge National Nature Reserve, which was included on the Ramsar list, have degraded by nearly 30%. Wetland degradation is a long-term continuous process with annual or interannual changes in water area, water level, or vegetation presence and growth. Therefore, it requires sufficiently frequent and high-spatial-resolution data to represent its dynamics. This study mapped yearly land-use maps with 30-m resolution from 1985 to 2018 using Landsat data in Google Earth Engine (GEE) to explore the wetland degradation process and mapped 12-day interval land-use maps with 15-m resolution using the Sentinel-1B and Sentinel-2 data in GEE and other assistant platforms to study the characteristics of wetland dynamics in 2018. Four sets of maps were generated using Sentinel-1B (S1), Sentinel-2 (S2), the combination of Sentinel-1B and Sentinel-2 (S12), and S12 with multitemporal remote sensing (S12′). All of the classifications were performed in the Random Forest Classification (RFC) method using remote sensing indicators. The results indicate that S12′ was the most accurate. Then, the impact of the historic land-use degradation process on current wetland change dynamics was discussed. Stable, degradation, and restoration periods were identified according to the annual changes in wetlands. The degraded, stable, restored, and vulnerable zones were assessed based on the transformation characteristics among wetlands and other land-use types. The impact of historical land-use trajectories on wetland change characteristics nowadays is diverse in land-use types and distributions, and the ecological environment quality is the comprehensive result of the effect of historical land-use trajectories and the amount of rainfall and receding water from paddy fields. This study offers a new method to map high-spatiotemporal-resolution land-use (S12′) and addresses the relationship between historic wetland change characteristics and its status quo. The findings are also applicable to wetland research in other regions. This study could provide more detailed scientific guidance for wetland managers by quickly detecting wetland changes at a finer spatiotemporal resolution.

**Keywords:** wetland loss; synthetic aperture radar; optical remote sensing; random forest classification; land-use classification; multitemporal remote sensing

## 1. Introduction

Wetlands, which are permanently or intermittently watered and inundated or saturated lands [1], are the most essential type of ecosystem and play significant roles in water quality and quantity regulation [2], wildlife habitat provision [3], and climate change mitigation [4]. However, due to the intensive development of agriculture and industries, as well as changes in precipitation and flooding patterns or other climate conditions, wetlands have been undergoing degradation throughout the world [5,6]. Although numerous studies on wetland change and wetland degradation have been documented [7–9], previous studies have not focused on the impact of wetland change trajectory or wetland degradation process on current wetland pattern [10]. Previous studies usually had a spatial resolution of more than 30 m or yearly temporal resolution, which could not capture the highly dynamic nature of wetlands [11,12], which changed seasonally or intermittently (tens of days). Quantifying long-term-scale wetlands with high spatiotemporal resolution is fundamental for understanding wetland degradation processes and the relationship between historical wetland change and the status quo.

Although obtaining timely accurate information is a challenging task, people have put considerable efforts toward generating and updating detailed wetland inventories [13–15]. Field investigations are time-consuming, laborious, costly, and not always located in accessible areas. Remote sensing, a convenient and efficient monitoring method of wetlands, is widely used for fine temporal sampling over large and inaccessible regions [16,17]. Existing remote sensing methods include two types: optical and microwave. Long time series with high spatiotemporal resolution are the advantages of optical remote sensing, yet the poor effect in cloud cover and night conditions are its inherent disadvantage [18,19]. In contrast to optical systems, Synthetic Aperture Radar (SAR), an active microwave sensor, is independent of weather conditions and solar illumination and can even penetrate vegetation canopies to map the understory, but some datasets are expensive and may not support long-term dynamic monitoring [20]. Wetland degradation is a long-term continuous process with interannually dramatic dynamics, and research on the fine resolution of spatiotemporal monitoring (monthly or even shorter) in long time series is prevailing [21,22]. However, it is difficult to collect sufficient cloud-free optical data in that case, so the combination of optical and SAR data is promising [12,23]. In land-use classifications, optical indices, such as the nominalized difference vegetation index (NDVI) and normalized difference water index (NDWI), are commonly used [24]. For SAR data, polarimetric decomposition techniques are a potential method for mapping wetlands by X and L bands [25], but C-bands, such as free available Sentinel data, are seldom used [26].

Wetland mapping on a long-time scale along with high spatial resolution requires collecting and storing large datasets, and often involves complicated processing and manipulation. It is infeasible or time-consuming to use conventional image processing software, such as ENVI and ERDAS. The cloud-based platform Google Earth Engine (GEE, https://earthengine.google.com, accessed on 1 October 2021) is an open-access platform hosting petabyte-scale remote sensing datasets and can integrate with many machine learning tools, such as the Random Forest Classification (RFC) method [27], which help facilitate high spatiotemporal resolution land-use mapping. Many studies have demonstrated that land-use mapping performed well in GEE [28,29], even on a global scale [30,31]. However, global-scale land-use products may not be suitable for small-scale research, particularly in areas with special land-use types, as land-use types of global-scale products are relatively simple and universal.

Zhenlai County, Jilin Province, China, has diverse land-use types and rich wetland resources, so wetlands in this territory were recognized as Wetlands of International Importance by the Ramsar Convention. Zhenlai has a special but locally common land-use type, saline-alkali land, which is not included in global-scale land-use type products. Previous wetland products often have different datasets and methods, as well as inconsistencies in spatiotemporal resolution [32–34], and their evaluation and comparison are difficult. Recent studies [35,36] have a trend of long-term series study, such as Cui [37], who explored

the causes of wetland landscape patterns in Momoge National Nature Reserve from 1984 to 2018 based on the GEE Platform with Landsat data, yet failed to consider interannual wetland dynamic changes. These studies are not enough to understand the process of wetland degradation or the relationship between wetlands both currently and historically.

To better explore the wetland degradation process and its influence on current wetlands, taking Zhenlai County as an example, this study was divided into three parts. (1) Land-use maps were generated annually from 1985 to 2018 and every 12 days in 2018 on the GEE and other assistant platforms. In the processing of the 12-day interval land-use maps, four sets of maps were made in Sentinel-1B (S1), Sentinel-2 (S2), the combination of Sentinel-1B and Sentinel-2 (S12), and S12 with the multitemporal remote sensing method (S12′). A fast and accurate land use classification process with a high spatiotemporal resolution could be found by comparing the four sets of maps. (2) The wetland loss process from 1985–2018 was evaluated. (3) The impact of historical wetland change trajectories on wetland change characteristics in 2018 was analyzed.

## 2. Study Area and Data

### 2.1. Study Area

Zhenlai County (N 45°28′14.3′′–N 46°18′15.8′′, E 122°47′6.3′′–E 124°04′33.7′′) is located in northwest Jilin Province, Northeast China (Figure 1), covering an area of 5350 km², bordering Heilongjiang Province to the east and Inner Mongolia to the west. As a part of the agro-pastoral transitional zone in the western Songnen Plain, the eco-environment of this area is very fragile to disturbances from human activities and natural impacts, such as farmland irrigation, overgrazing, and extreme rainfall or drought. The typical wetland types are saline-sodic wetlands and water bodies largely located in the Momoge National Nature Reserve (1440 km²), which was included on the List of Wetlands of International Importance (Ramsar sites) in 2013 as the main stopover site for 90% of the world's Siberian Cranes. The region experiences a temperate continental monsoon climate, with an annual evaporation of 1489 mm and annual precipitation of 397 mm and 89% distributed from May to September. Low precipitation and high evaporation mainly result in soil salinization. The coldest and warmest temperature is approximately −16 °C in January and 23 °C in July. The frost-free period is 137 days. The main soil types are meadow soil and marsh soil.

### 2.2. Data and Platform

The data used in this study include remote sensing data, digital elevation model (DEM) data, validation data, and precipitation data. The optical remote sensing data from Landsat 5, 7, and 8, were used to map the 30-m-resolution yearly land-use map from 1985 to 2018 in the GEE platform. The SAR data Sentinel-1 and optical remote sensing data Sentinel-2 were selected to map the 15-m-resolution land-use trajectory map per 12 days in 2018 in GEE, ENVI, SARscape (https://www.sarmap.ch, accessed on 3 November 2021), and PolSARpro (https://step.esa.int, accessed on 3 November 2021). The data analysis and mapping were carried out in ArcGIS and R platforms.

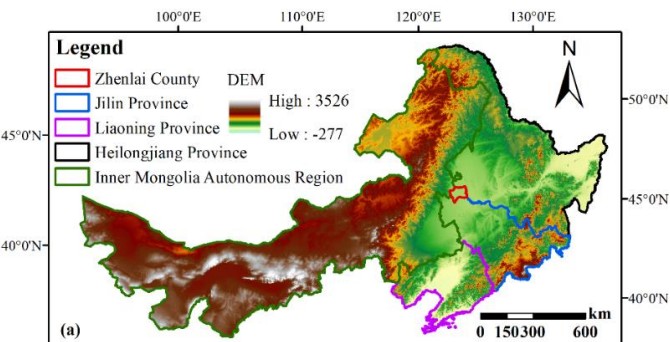

**Figure 1.** *Cont.*

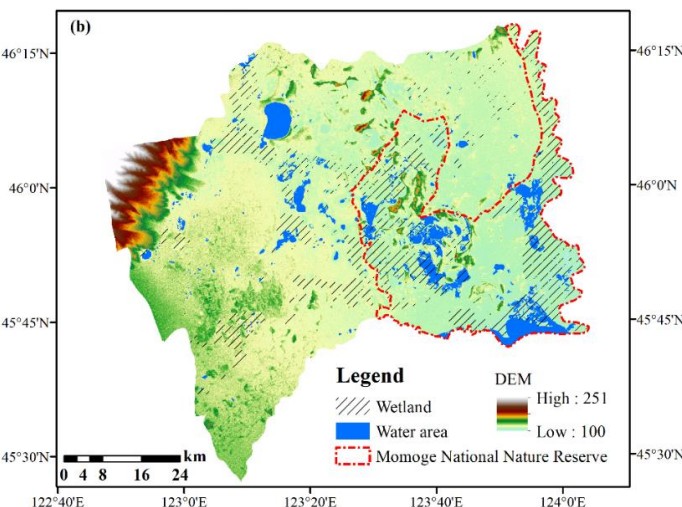

**Figure 1.** The location of Zhenlai County (red outline) in the four provinces of northeast China (**a**) and the wetland, water area, and reserve in Zhanlai County (**b**).

2.2.1. Remote Sensing Data

1.    Landsat

Landsat images have been widely used in land-use maps as they are open access, medium resolution (30 m), and have a short repeat cycle (16 days) [36,38]. Landsat surface reflectance data were used to map the yearly land-use map during the period 1985–2018 via the GEE platform in this study [28,39–41]. The study area was covered by the satellite track of row 28, paths 119, 120, and 121. A total of 2562 Landsat images were used during this period, and the details in each year are shown in the supplementary material (Table S2).

2.    Sentinel-1

The Sentinel-1 satellite (https://scihub.copernicus.eu, accessed on 3 November 2021) is comprised of a constellation of two polar-orbiting satellites (Sentinel-1A and Sentinel-1B) that were launched in 2014 (Sentinel-1A) and 2016 (Sentinel-1B) by the European Space Agency, operating day and night and performing C-band SAR images [42]. Due to its short revisit time (6–12 days), fine resolution (down to 5 m), large coverage (up to 400 km), and cost-free availability, it has been widely used in land cover identification studies in recent years [43,44]. Fifteen Sentinel-1B (wavelength λ: 5.6 cm, incidence angle θ: 38.78°) images in interferometric wide swath mode (IW) single look complex (SLC) dual polarimetry (VV+VH) format were acquired over the area, spanning from 29 April to 14 October 2018, at a twelve-day interval. These images were used to generate backscatter, coherence, and interferogram parameters for land-use mapping, especially for exploring the wetland change trajectory in 2018 [45].

3.    Sentinel-2

Similar to Sentinel-1, Sentinel-2 also has two satellites, Sentinel-2A (launched in June 2015) and Sentinel-2B (July 2017). Sentinel-2 is a wide-swath, high-resolution, high revisit frequency (2–5 days), optical imaging mission. It has 13 spectral bands with four bands at 10 m, six at 20 m, and three at 60 m spatial resolution. In this study, the 10-m-resolution bands blue (B2, 0.490 μm), green (B3, 0.560 μm), red (B4, 0.665 μm), and near-infrared (NIR, B8, 0.842 μm) and the 20-m-resolution band short-wavelength infrared (SWIR, B11, 1.610 μm) were used to interpret the land-use types in 2018 [46].

Sentinel-2 Top of Atmosphere reflectance data was used in the GEE platform. The Sentinel-2 images that were near the acquisition time of Sentinel-1 data were selected to compose a cloud-free or near-cloud-free image. A QA60 band, indicating the amount of opaque and cirrus clouds computed based on spectral criteria, was used to exclude the

invalid pixels in each Sentinel-2 image. Finally, 158 Sentinel-2A and Sentinel-2B images covered by orbits 89 and 132 were selected in this study.

### 2.2.2. Other Data

1.  Training and validation data

A total of more than 500 sites were chosen as regions of interest (ROIs) by field investigation in May, June, and September of 2018. The ROIs of other periods (other months in 2018 and the past years from 1985 to 2017) were modified based on the field data through high spatial resolution images accessed in Google Earth and talking with local landowners. The ROI for each map was randomly divided into 70% for training data and 30% for validation data.

2.  DEM

The DEM with a spatial resolution of 12.5 m was obtained from the Alaska Satellite Facility (ASF, https://www.asf.alaska.edu/, accessed on 3 November 2021), which was derived from the Advanced Land Observing Satellite Phased Array Type L-band Synthetic Aperture Radar (ALOS PALSAR) imagery. The DEM was used to calculate the coherence coefficient, backscatter coefficient, and polarimetric decomposition process with Sentinel-1 data (Figure 2).

3.  Precipitation data

The daily precipitation datasets from 1 January 1985 to 31 December 2018 were downloaded from the China Meteorological Data Service Centre (CMDC, http://data.cma.cn, accessed on 3 November 2021). The mean value of Tailai station and Baicheng station was used as the final precipitation data in Zhenlai County.

## 3. Methods

There were two parts to evaluate the wetland change trajectory in Zhenlai County. First, yearly land-use maps during the period 1985–2018 were built to assess wetland loss (the blue parts in Figure 2). Then, the land-use maps every 12 days from 29 April to 14 October 2018, were mapped to reflect the characteristics of wetland change and to evaluate the impact of historical wetland loss on wetland change characteristics in 2018 (the green parts in Figure 2). All classifications were performed with the Random Forest Classification (RFC) method on the GEE or ENVI platform. RFC is an ensemble learning method. It is based on a large number of decision trees with the training data, and splits the nodes by minimizing the correlation between trees, and assigns a land-use type to each pixel on the basis of the majority vote of trees [47,48].

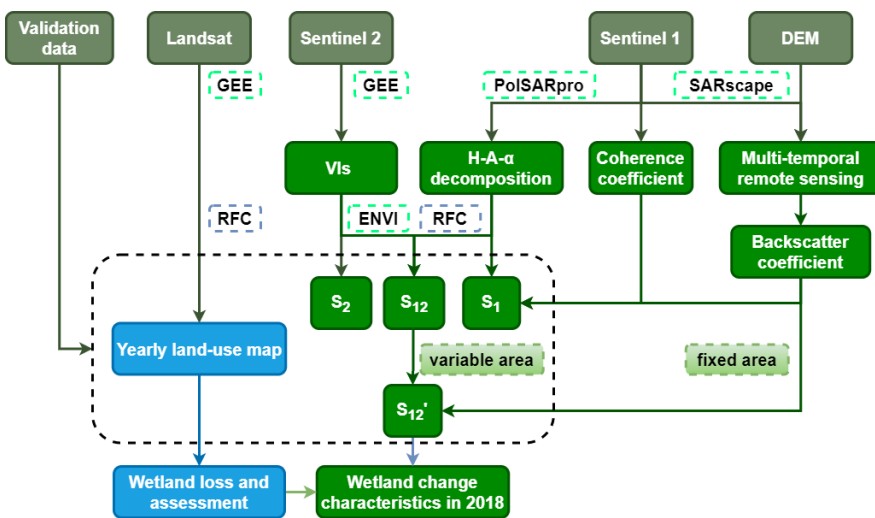

**Figure 2.** Research methods and analytical framework.

### 3.1. Yearly Land-Use Mapping in GEE

Multitemporal Landsat 5, 7, and 8 images spanning 32 years were used to show the spatiotemporal variability of land-use dynamics, especially for wetland change trajectories. The QA bitmask band was also used to mask out bad-quality pixels caused by opaque and cirrus clouds in each image. Before classification, eight land cover types were defined for the study area based on the field investigation, including construction land, dry farmland, forestland, grassland, paddy field, saline-alkali land, water area, and marsh. For better observation and understanding of the wetland change characteristics, marsh is hereafter referred to as inundated or saturated land and water area as water body extent, wetland contained marsh and water area. The spectral index was sensitive to different land-use types [49]. In this study, five spectral indices, including the nominalized difference vegetation index (NDVI) [50], enhanced vegetation index (EVI) [51], land surface water index (LSWI) [52], normalized difference water index (NDWI) [53], and modified normalized difference water index (mNDWI) [54], were calculated by the 30-m-resolution bands (blue, green, red, NIR, and SWIR) of Landsat in GEE. See the detailed calculation process in Table S1. NDVI and EVI were used to estimate the degree of vegetation coverage in terms of vegetation height, density, and biomass. LSWI, NDWI, and mNDWI could differentiate water bodies and wetness surfaces (normally with positive values) from the background (trending to negative values). Due to the high temporal dynamics of marsh, becoming grassland during the dry season and water area during the flood season, the maximum, minimum, and mean of each index in a year were generated to compose a new image collection.

The new image collections with 15 bands and the ROIs in each year during 1985–2018 were input into the RFC model to discriminate the land-use types. The ROIs for each map were randomly divided into 70% for training data and 30% for validation data. Overall accuracy (0–1) and the kappa index (0–1) were calculated to test the accuracy of the classification results. Overall accuracy was measured by dividing the total number of correctly classified samples by the total number of validation samples. The kappa index indicated the consistency between the interpretation data and the validation data [48,55]. The higher the two values were, the more accurate the classifications. All the classification tasks were conducted in the GEE platform.

### 3.2. Analyzing Interannual Variation of Wetland in 2018

The 12-day SAR Sentinel-1B and 5-day optical Sentinel-2 data provided an opportunity to form high spatiotemporal resolution land-use maps. To better explore the dynamics of marsh and water areas, water areas were divided into shallow-water area and deep-water area. The remaining land-use types were the same as the land-use types mentioned above in the Landsat interpretation. The temporal resolution (12 days) was chosen for Sentinel-1B data in this study, and Sentinel-2 images that were around the acquisition time of Sentinel-1 data were selected in medium value to compose a cloud-free or near-cloud-free image in GEE.

Three sets of land-use types were mapped, first: S1 (using only the Sentinel-1B data), S2 (using only the Sentinel-2 data), and S12 (using the composite of Sentinel-1B and Sentinel-2 data). To fill the holes in the Sentinel-2 image when removing cloudy pixels through the QA60 band, especially for cloudy and rainy days from June to September, SAR Sentinel-1B, which is independent of solar radiation, could complement Sentinel-2 under cloudy and rainy weather conditions. Second, it was postulated that the S12 set would perform better than S1 and S2, and we regarded it as a base map to make further improvements. The "fixed area" map was mapped using multitemporal analysis with Sentinel-1B data. The "fixed area" is the zone where the land-use area does not change within a year. The combination of "fixed area" and S12 led to another land use set named S12′. Finally, four sets (S1, S2, S12, and S12′) with fourteen land-use maps for each set were resampled to a spatial resolution of 15 m and temporal resolution of 12 days spanning from

29 April to 14 October 2018. The land-use classifications of S1, S2, S12, and S12 were all performed in the RFC model of ENVI.

In this study, 15 parameters (12 in the polarimetric decomposition method) of Sentinel-1B were generated concerning their potential for monitoring land-use change. SARscape, a modular set of ENVI software, was used to generate the backscatter coefficient ($\sigma^0$, dB) and coherence coefficient (CC) value of Sentinel-1B images. $\sigma^0$ is the square of the amplitude with complex signals recorded by the radar instrument [56]. Sentinel-1B is a dual-polarimetric (VV+VH) SAR data, so $\sigma^0$ was in VV polarization and VH polarization, which enables a more accurate separation of the land-use structure. There were three backscattering mechanisms among different land-use types: double-bounce, volume, and surface scattering, and the value of $\sigma^0$ decreased in this order. Therefore, $\sigma^0$ can be considered an indicator of surface type changes. The coherence coefficient (CC) value of two adjacent images is another parameter often employed in land-use identification and classification [57,58]. Coherence is the magnitude of an interferogram's pixels, divided by the product of the magnitudes of the original image's pixels. It depends on the $\sigma^0$ value and the changes between the acquisition time of two adjacent images, ranging from 0 (no useful information) to 1.0 (no noise). Coherence can serve as a measure of the land-use type and can identify when a tiny, otherwise invisible change (electromagnetic behavior) has occurred in the images. Building $\sigma^0$ and CC consists of five steps: (1) generating the 'four range and one azimuth' multilook intensity images (14.88 m in range resolution and 13.88 m in azimuth resolution) to enhance the radiometric resolution of the radar signal and the signal/noise ratio. (2) Coregistering all the Sentinel-1B images together with the reference DEM. (3) Filtering through a De Grandi Spatiotemporal Filter ($\sigma^0$) and Goldstein filter (CC) to decrease the speckle. (4) Generating the interferogram and derive the CC of two adjacent images. (5) Using the range-Doppler approach for geocoding and radiometric calibration, and resampling the images into normalized $\sigma^0$ values in decibels with 15 m resolution [59].

Land-use types differ in their surface-scattering behaviors due to the differentiation of structure and texture (roughness and wetness, etc.). The polarimetric decomposition technique aims to separate the polarimetric signals physically into different scattering mechanisms (surface, double-bounce, and volume scattering) to improve the classification with Sentinel-1B data. Entropy/anisotropy/alpha (H/A/$\alpha$) decomposition [60] was used in this study with PolSARpro software. It was based on the eigenvalues and eigenvectors of the coherency metric, and the parameters of entropy, alpha angle, anisotropy, and their derivatives (Shannon entropy and the combinations of entropy and anisotropy) were calculated (Table S1, Figure 3). The alpha angle ($\alpha$, 0~90°) describes the dominant type of backscattering, the entropy (H) from 0 to 1 is indicative of a scattering mechanism from a major to the random mixture, and anisotropy (A) is the normalized eigenvalue difference indicating whether multiple scattering mechanisms are occurring [61]. The new image collection formed by 15 parameters was used to generate land-use map S1.

The medium values of optical parameters (NDVI, EVI, NDWI, mNDWI, and LSWI) of Sentinel-2 images that were near the acquisition time of Sentinel-1 data were first collected to map S2 in the GEE platform and classified in ENVI. Nineteen parameters (14 in Sentinel-1B and 5 in Sentinel-2 data), many of which are highly correlated (Figure 3), were composed to map S12.

The "fixed area" was mapped by the $\sigma^0$ of multitemporal Sentinel-1B data, and the "variable area" map was based on S12. The division of "fixed area" and "variable area" primarily depended on whether the land-use area changes within a year. The "fixed area" includes construction land, dry farmland, forestland, and paddy field, and the variable area includes grassland, saline-alkali land, water area, and marsh. If the "fixed area" land-use types emerge in the "variable area" map, they will be exchanged by similar classes: construction land would be exchanged by saline-alkali land, dry farmland exchanged by grassland, and forestland and paddy field exchanged by marsh. Then, the twelve-day land-

use map S12' was built with optical (Sentinel-2) and SAR (Sentinel-1) data in polarimetric decomposition and a multitemporal method.

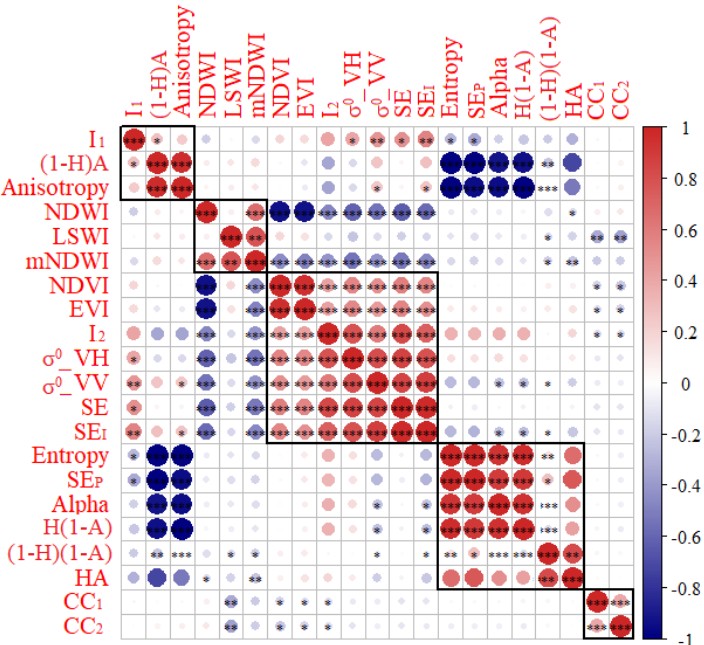

**Figure 3.** The correlation coefficient heatmap of parameters derived from the Sentinel sensor (CC1 and CC2 are the CC values of the previous period and the next period, respectively, with only CC1 on 29 April and CC2 on 14 October). Additional details on the calculation of parameters are provided in the Supplemental Materials (Table S1).

### 3.3. The Impact of Historical Wetland Change Patterns on Current Wetland Characteristics

Wetland loss assessment was based on the spatiotemporal wetland change characteristics: dividing into three periods based on the wetland change trajectory, redefining the wetland type in each period based on the occurrence frequency of wetland in the counterpart period, and evaluating comprehensively based on the change characteristics of redefined wetland in three periods.

According to the transformation characteristics among wetlands and other land-use types, the study time (1985–2018) was divided into three periods: stable period (1985–1994), degradation period (1995–2010), and restoration period (2011–2018). The redefinitions of marsh and water area were calculated by the frequencies of marsh and water area in the counterpart period. If the raster value was more than 80% in the water area or marsh in this period, it was defined as wetland, and the remains were the other land-use type. In the wetland, if more than 80% are in the water area, the raster would be defined as the water area, and the remains would be marsh. The marsh and water areas in the corresponding periods were named SW (stable water area), SM (stable marsh), DW (degraded water area), DM (degraded marsh), RW (restored water area), and RM (restored marsh). Using the frequency of wetland occurrence at a certain stage instead of the raster value of one year could effectively reduce wetland assessment errors caused by interpretation errors. Eight assessment zones were divided by historical wetland dynamic characteristics (Table 1).

We performed zonal statistics on each land use map in 2018 using ArcGIS so that the area of various land use types in 2018 was obtained for each assessment zone. Dynamic changes in land use types among the 8 zones were compared to infer the influence of historical land use change on current wetland dynamics.

**Table 1.** The determination of evaluation zones based on wetland dynamics.

| Evaluation Zone | Calculation |
|---|---|
| MSA (Marsh stabilization area) | SM ∩ DM ∩ RM |
| WSA (Water stabilization area) | SW ∩ DW ∩ RW |
| MDA (Marsh degradation area) | SM ∩ RO |
| WDA (Water degradation area) | SW ∩ (RO ∪ RM) |
| MRA (Marsh restoration area) | SM ∩ (RM ∪ RW)-MSA-WSA |
| WRA (Water restoration area) | SW ∩ RW-MSA-WSA |
| MVA (Marsh vulnerable area) | SM50% ∩ DM50% ∩ RM50%-MSA-WSA-MDA-WDA-MRA-WRA |
| WVA (Water vulnerable area) | SW50% ∩ DW50% ∩ RW50%-MSA-WSA-MDA-WDA-MRA-WRA |

50% means the raster value of marsh or water area is more than 50% instead of 80%.

## 4. Results

### 4.1. Yearly Land-Use Mapping in GEE

The overall accuracies and kappa indices of classification were all over 80% from 1985 to 2018 (Table S2), indicating a rapid and efficient interpretation process based on the vegetation and water parameters with RFC in the GEE platform (Figure 4). The annual area of land-use types are shown in Figure S1. Dry farmland (28.29%) and grassland (27.19%) are typically found in Zhenlai County, and the annual average areas of water and marsh were 309.33 km$^2$ and 962.29 km$^2$, respectively. The extent of water area and marsh changed dramatically during the thirty-four years, which was attributed to the complex interactions among human development, regulation of water conservancy facilities, and climatic change characteristics.

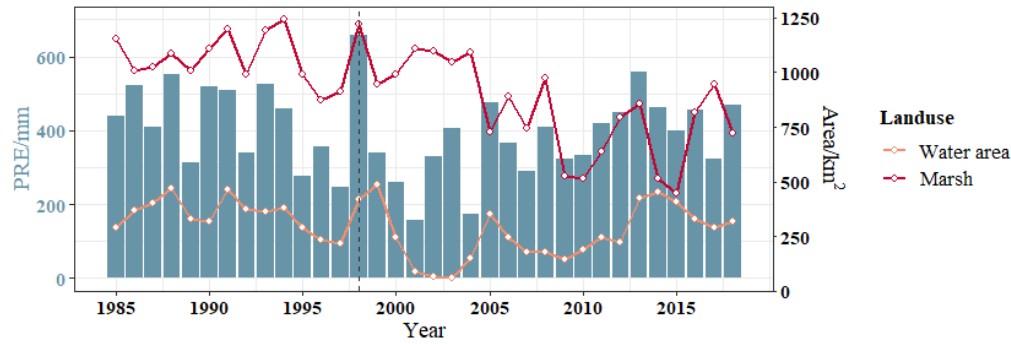

**Figure 4.** Characteristics of water area, marsh, and precipitation from 1985 to 2018.

During 1985–1994, the areas of water and marsh fluctuated slightly due to rainfall and were stably maintained at 390.92 km$^2$ and 1125.87 km$^2$, respectively. From 1995 to 2009 (except for 1998), the area declined turbulently due to human development. Important flood events took place in 1998, and precipitation reached 658 mm, after which the areas of marshes and water bodies reached their highest values of 1345.87 km$^2$ and 499.18 km$^2$, respectively. Then, a drying period lasted for three years, and the area of water reached its lowest value of 75.47 km$^2$. After 2009, the area was mainly influenced by precipitation, especially in the years around 2015. There was a trade-off between water area and marsh along with precipitation, and the minimum marsh area was 480.59 km$^2$.

### 4.2. Wetland Change Characteristics in 2018

#### 4.2.1. Twelve Days of Land-Use Mapping with Optical and SAR Data

The interpretation differences based on different data sources or methods have been shown in Figure 5. Confusion errors occurred among all interpretations, especially in the single use of Sentinel-1 and Sentinel-2 data (S1 and S2). The classification using the combination of Sentinel-1 and Sentinel-2 data (S12) was more accurate, but not obvious, and the inclusion of multitemporal analysis significantly improved the classification accuracy, especially for wetlands (Figure S2). The yellow windows in S1 and S12 mistook the marsh

as other land-use types. Even though a variety of polarimetry decomposition indices were used for detecting the land-use types, marshes were still easily misclassified to others as they shared a similar spectral profile or the same scattering mechanism. Confusion was also common due to the shadow influence, such as the black window in S2 and S12. In the green windows of S1, S2, and S12, the grassland was erroneously identified as marsh or water areas. This may be attributed to the incomprehensiveness of the optical and SAR parameters or other noises.

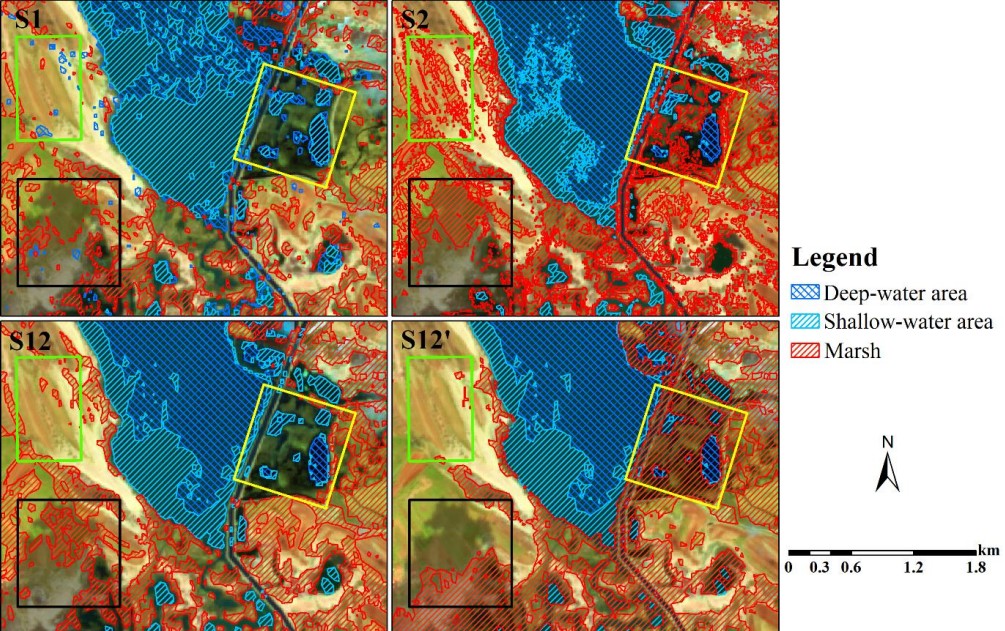

**Figure 5.** The interpretation differences based on different data sources or methods. S1 was the land-use map based on Sentinel-1 data using polarization decomposition and the D-InSAR method, S2 was based on Sentinel-2 data using the vegetation and water indices, S12 was the combination of S1 and S2, and S12′ was based on S12 and the multitemporal remote sensing method on Sentinel-1B data. There were three types of misclassifications: shadows in the black window, other land-use types in the green window, and missing marshes in the yellow window.

To better compare the interpretation results, the kappa index is shown in Figure 6. The ROIs used to calculate the kappa index were selected where land-use type did not change over time. The classification accuracy over all periods (multigroup kappa index) was highest in S12′ and lowest in S2, indicating that even though a variety of optical indices for the detection of vegetation and water were performed, the classification faced many error identifications (Figures 5, S2 and S12). The kappa value of S12 was higher than those of S1 and S2, showing the complementary advantage of optical and SAR indices. In particular, an improvement of approximately 13% in the kappa index of S12′ compared to S12, illustrated that the method of identifying and replacing land-use types with fixed areas in S12 based on multitemporal analysis was an efficient classification method.

In this study, the importance information was used: increase in mean square error% (%IncMSE) and increase in node purity (IncNodePurity) to evaluate different parameters for the classification results (Figure 7). A comparison of the kappa index and the importance information showed that the optical land-use map (S2) was less accurate than SAR (S1), but the optical indices were more important than radar. This suggests that different input indices in the RFC method contain different characteristics of ground targets, which may play different roles in the classification results. The vegetation and water conditions were directly captured by optical parameters, such as NDVI and NDWI, and the SAR indices, especially $\sigma^0$ and alpha, were good at identifying the geometric characteristics (height, orientation, and shape) of land-use types. Accordingly, it was concluded that the

complement of optical and SAR data was very useful for improving classification accuracy, and incorporation with multitemporal analysis could enhance the discrimination of land use types.

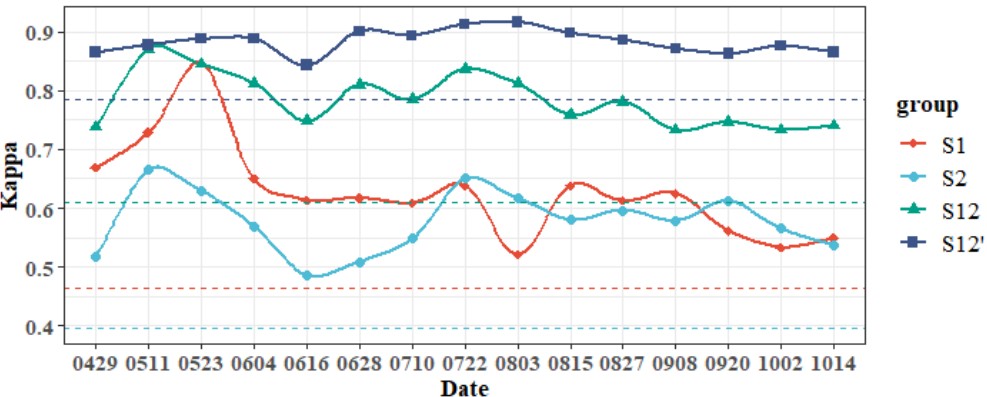

**Figure 6.** Kappa index of land-use maps coming from different data sources or methods in 2018. The dashed line in each color indicates the multigroup kappa index at all times.

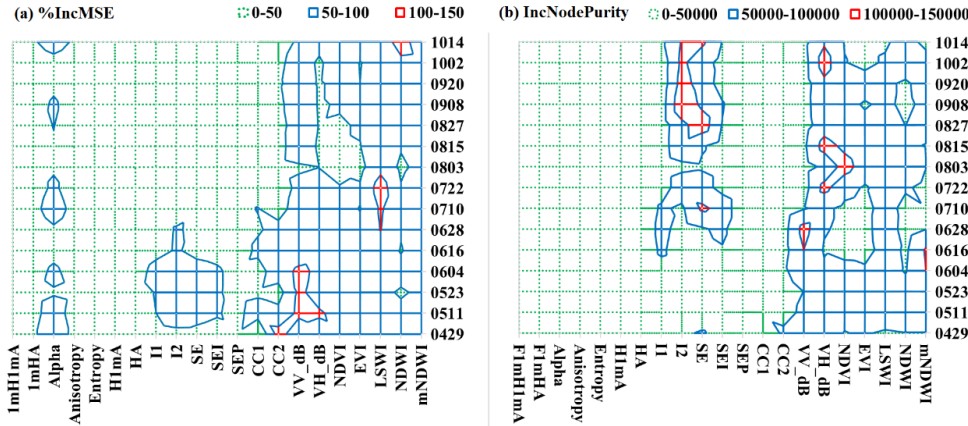

**Figure 7.** Importance values: (**a**) increase in mean square error% and (**b**) increase in node purity of the individual parameters in land-use mapping based on the RFC method.

### 4.2.2. The Parameter Characteristics in Different Land-Use Types

The NDVI from optical data and backscattering coefficient ($\sigma^0$) from SAR data had a significant contribution to the classification of all the parameters (Figure 8). NDVI and $\sigma^0$ had different temporal behaviors in different land-use types, which also indicated accurate classification results. The results reported herein were consistent with those reported in previous studies. The NDVI was less than 0 in the deep- and shallow-water areas, from 0 to 0.2 in the bare saline-alkali land and construction, 0.2 to 0.4 in saline-alkali land and the mixed bare saline-alkali land and vegetation, and larger than 0.4 was covered by vegetation. NDVI could detect the characterization of vegetation phenology and peaked in July and August when vegetation growth was most vigorous.

Land use classes and changes can be identified by analyzing the temporal behavior of $\sigma^0$. The backscatter from the water area was very low due to specular reflection, and very little backscatter was returned to the sensor. $\sigma^0$ in the deep-water area was higher than that in the shallow-water area due to the surface reflector by waves caused by wind. $\sigma^0$ was highest in the construction and forest areas due to more double-bounce scattering. $\sigma^0$ from farmland was higher than that from the water area and increased with the complex interaction between crops and grounds caused by irrigation in June. $\sigma^0$ was higher in wet soil, such as in marshes, varying between $-11.61$ and $-8.47$ dB. The changes in the density of the plant canopy and water level could elect different or multiple scattering

mechanisms. The increase from April to August might be explained by the more double bounce backscattering of the radar waves between growing stalks and the water surface and some volume scattering in the vegetation canopy. The decrease in the leaf-off season (September and October) was due to the combined impact of progressive putrefaction and dry plants with minor double bounce specular plants.

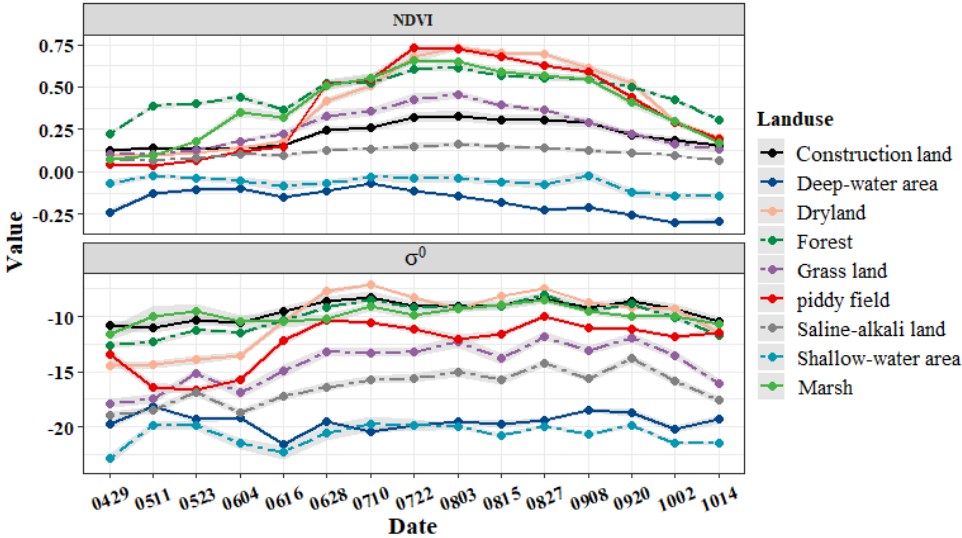

**Figure 8.** The temporal characteristics of NDVI and $\sigma^0$ in different land-use types. The shaded area of each line represents the 95% confidence interval of the standard error.

### 4.3. The Impact of Historical Wetland Changes on Current Wetland Dynamics

#### 4.3.1. Wetland Loss Assessment

From 1985 to 2018, the marsh area decreased by 36.20%, and major transitions occurred in farmland (dry farmland and paddy field, 18.46% and 23.32%), grassland (27.95%), and water area (15.16%). According to the characteristics of marsh transformation, three marsh transformation periods were divided (Figure 9). The stable period was from 1985 to 1994, and the areas of marsh trans-from or trans-to were small and similar. The degradation period was 1995–2010, and the extent of marshes declined obviously. The marshes in 1997–2000 were mainly affected by the flood event that took place in 1998. Before 1997 and after 2000 in the degradation period, marsh output was larger than input, and the marsh transformation pattern changed dramatically after 2000, particularly in the transformation between marsh and farmland, indicating that human development increased at that time. Marshes have been undergoing restoration since 2011, and the area transferred to farmland has decreased. The high value of marsh transferred to W&G&S (water area, grassland, and saline-alkali) may be due to the decreasing precipitation during 2013–2015, and the reason for the decrease in marsh input and the increase in output after 2016 needs to be further studied.

According to the transformation characteristics among wetlands and other land-use types, wetland assessment was divided into eight categories: degraded, stable, restored, and vulnerable zones of water area and marsh (Table 1, Figure 10), and the area of each zone decreased in this order. Water stabilization areas (WSAs, 96.49 km²) were mainly distributed in larger reservoirs, lakes, and rivers, marsh stabilization areas (MSAs, 272.63 km²) were mainly distributed in the banks of Nenjiang, the river in eastern Zhenlai County. The areas that had been restored were near the stabilization zones, with a total area of 212.12 km². The vulnerable area was small (36.63 km²) and scattered, and the vulnerable area in water could be ignored. The degraded area was the largest area among the different evaluation zones, the water degradation areas (WDAs, 127.60 km²) were mainly distributed in small pools, and the marsh degradation areas (MDAs, 642.85 km²) were the largest and could be used as the corridor area connecting marshes, pools, rivers, and lakes of Zhenlai

County. Currently, MDAs are primarily river flooded grasslands from the northwest to the middle of Zhenlai County and dry farmland in the south.

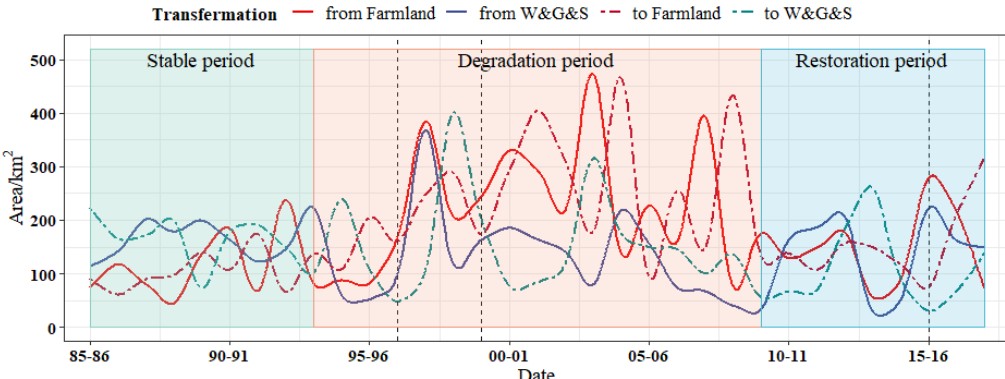

**Figure 9.** Marsh transformation in different periods, from or to W&G&S indicating the marsh trans-from or trans-to the water area, grassland, and saline-alkali; the dashed line is the internal segmentation in degradation period and restoration period.

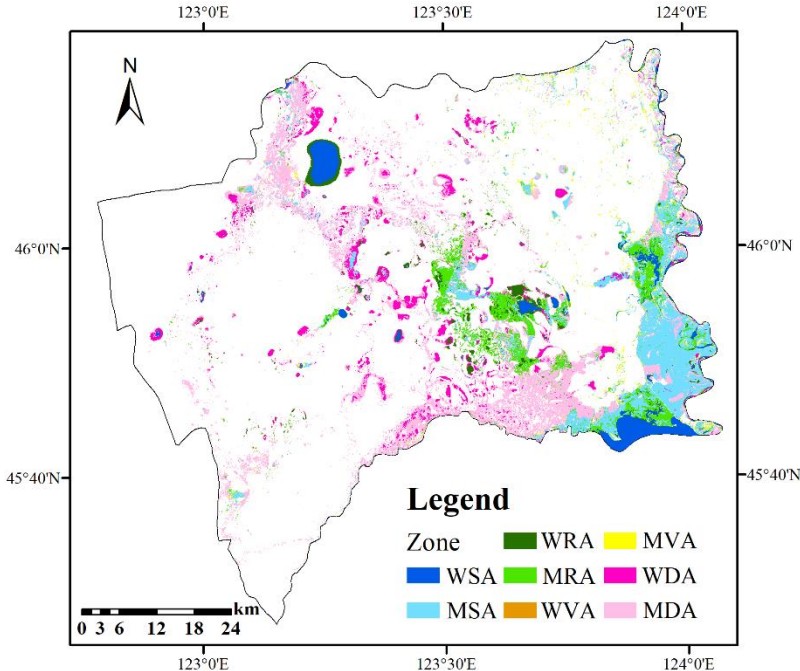

**Figure 10.** Wetland (marsh and water area) loss evaluation.

4.3.2. The Impact of Historical Wetland Changes on Current Wetland Dynamics

The historical changes in marsh and water areas have had an impact on their temporal characteristics (Figure 11). There was more deep-water area in the WSA, more shallow-water area in the WRA and marsh emerged in this zone before August. Saline-alkali land and farmland dominated in WDA, with both deep and shallow water accounting for a small proportion. The area of WVA was the smallest, but the transition was the most dramatic between the deep-water area and shallow-water area. The ratio of marsh area to all land-use types in MSA, MRA, and MVA was larger than that in MDA, but the area of marsh was smallest in MVA. The MVA and MDA had large ratio of farmland. In the MRA, wetland changes mainly occurred in marsh and shallow-water area, and the deep-water area changed little. The deep-water area, shallow-water area, and marsh were mostly influenced by precipitation and water receding from the paddy field. With the increase in precipitation and receding water, there was a tendency for marshes to transform to

shallow-water areas, and shallow-water areas to deep-water areas. The intensity of this trend decreased in the order of WVA, WSA, WRA, WDA, MSA, MRA, MVA, and MDA.

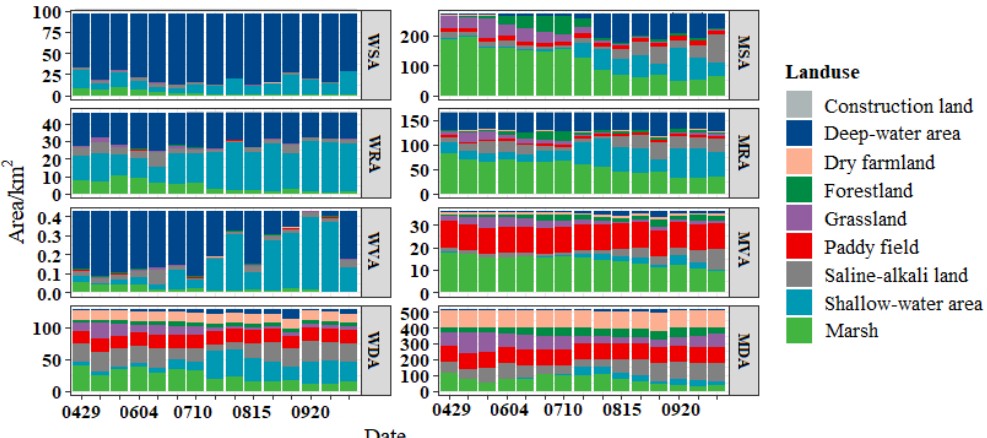

**Figure 11.** Temporal change characteristics of the land-use area in the different wetland loss assessment zones.

The quality of the ecological environment was distinct in various marsh and water area loss assessment zones (Figure 12). Three representative indices were selected to represent the temporal change characteristics in different assessment zones: $\sigma^0$, NDVI, and mNDWI. $\Sigma^0$ and NDVI had similar trends in different zones, increasing steadily to July and then decreasing until October. The marsh zones had higher values of both $\sigma^0$ and NDVI than the water area zones, except for ADA. The higher value in ADA was caused by the high percentage of paddy fields, grasslands, dry farmlands, and marshes. Before July, the NDVI was the highest in the ESA owing to the large portion of growing vegetation. A significant decrease in NDVI in July was due to the increase in water area, and a significant increase in mNDWI also proved this result. The steady increase in NDVI in EVA, EDA, and ADA was caused by the growth of crops. The more water areas there are in ASA, AVA, and ARA, the lower the $\sigma^0$ value due to the specular off of the radar signal. The higher $\sigma^0$ in EDA and EVA was caused by the regularly arranged crops, which had a higher backscatter coefficient. $\Sigma^0$, NDVI, and mNDWI have significantly influenced by farmland [62–64]. The more farmlands exist, the more complicated to estimate the wetland ecological environment among different wetland assessment zones.

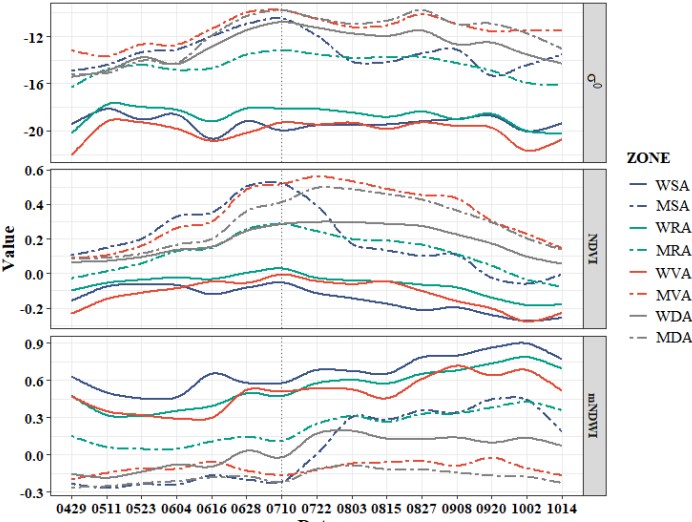

**Figure 12.** Temporal change characteristics of indices in different marsh and water area loss assessment zones.

## 5. Discussion

### 5.1. Wetland Mapping

#### 5.1.1. Quickly and Accurately Generating Annual Land-Use Maps in GEE

More than 80% of the pixels of land-use maps of each year during 1985–2018 were correctly identified in this study (Table S2), which was attributed to the sensitive indices of vegetation and water parameters, efficient method with RFC, and convenient platform on GEE. NDVI and EVI are sensitive to photosynthetically active biomass and are often used to estimate the fraction of vegetation coverage as tall or dense vegetation absorbs the most incident visible light and reflects a large portion of the NIR light [65]. LSWI, NDWI, and mNDWI are sensitive to water and wet surfaces due to the strong absorption in NIR and SWIR [66]. The maximum, minimum, and mean of each index within a year can represent the extent variation in water area and the phenological characterization of vegetation in a year [67]. The parameters were calculated by the ratio operation with various bands, which could decrease several multiplication noises, such as sun illumination differences, atmospheric attenuation, and topographic variations [48]. The convenience and superiority of the GEE platform and free sharing of Landsat and Sentinel images made this study possible to explore land-use dynamics with high spatiotemporal resolution. In addition to its fast computing and large storage capacity, many machine learning algorithms have been integrated into the JavaScript or Python application programming interface (API). RFC [68] is a nonparametric classifier that can handle high-dimensional datasets (more than 20 indices with 5 GB in this study) without preselection and overtraining of classification parameters, and the classifier is also relatively robust to noise and no data value, which is common in optical data due to removing cloudy pixels.

#### 5.1.2. High Spatiotemporal Land-Use Mapping

This study demonstrated an effective method in high-temporal-resolution land-use mapping: using the optical and SAR parameters in the RFC method and combining the multitemporal remote sensing method, which was used to identify the "fixed area". The feasibility of polarization-decomposed C-band data and the effectivity of the fusion of Sentinel-1 and Sentinel-2 data in land use classification were discussed. Four sets of land-use maps using the Sentinel-1 and Sentinel-2 data were derived: the single use of Sentinel-2 and Sentinel-1B data (S2 and S1), the combination of Sentinel-1 and Sentinel-2 data (S12), the inclusion of multitemporal analysis (S12′) after S12, and the classification accuracies increased in this order, which was similar to previous studies. An increase in the number of parameters generally led to an increase in accuracy [12]. Comparing the four sets of land-use maps (Figure 5), confusion with other land cover types and shadows was common in S1, S2, and S12. The misclassification of S1 was likely a consequence of the partially overlapping spectral profiles or scattering mechanisms among different land-use types [69,70]. Sentinel-1B is a dual-polarimetric (VV+VH) SAR data, and dual polarimetry is a polarimetric subspace of full or quad polarimetry (HH, HV, VH, and VV) [71,72], which limits the conduction of polarimetric analyses [73,74]. For example, the derivatives of entropy (H) and anisotropy (A) in full polarimetry (1-H)(1-A) indicate the presence of a single dominant scattering mechanism, AH and A(1-H), in the presence of two equally and unequally probable scattering mechanisms, and H(1-A) corresponds to random scattering. For dual polarimetry, the interpretation of H-A combinations was not clear. Therefore, the quantity and quality of scattering mechanisms involved in the scattering process are unambiguous [72]. Further studies to explore the difference between dual and full polarimetric SAR data in land-use mapping are essential. For S2, the main noise source was the no data value and shade caused by the cloud. The better performance of S12 shows the complementary advantage of optical and SAR indices, as several studies highlighted [75,76]. The density, height, and biomass of vegetation could be captured by NDVI and EVI. The water and wetness conditions captured by NDWI, mNDWI, and LSWI, and the geometric characteristics (height, orientation, and shape) and physical characteristics (roughness and dielectric constant) captured by the polarimetric decomposition

parameters. The inclusion of optical and SAR indices enhanced the discrimination of land-use classes [77,78]. The significant improvement of S12' was not surprising given that it not only concluded the complementary indices, but also incorporated multitemporal analysis, which was used to identify the "fixed area" in a year. The phenological characteristics vary with vegetation type (marsh, farmland, grassland, and forest), which was very useful for improving classification accuracy.

The optical land-use map was less accurate than SAR, but the optical indices are more important than radar. This contrary value between the kappa index and the RFC importance information of optical and SAR data was seldom compared in previous studies, which suggests that different input indices in the RFC method play different roles in the classification results, as mentioned above. Optical indices are the most intuitive and effective, and SAR indices provide assistance, particularly in the night and cloudy or rainy seasons. The optical indices are more efficient than SAR indices, which has also been proven in recent studies [41,48]. The NDVI from optical data and $\sigma^0$ from SAR data have a significant contribution to the classification of all the parameters. Classes and changes can be discriminated well by analyzing the temporal behavior of NDVI and $\sigma^0$, which also indicates accurate classification results.

Unfortunately, most of the processes of the 12-day interval land-use mapping were conducted in the conventional image processing software SARscape and PolSARpro, not in GEE, as the codes for preprocessing and polarimetric decomposition of SAR data were too complicated. However, it is very promising to implement all the steps in GEE with the professional help of coders, which means that fast and accurate land-use mapping is possible, even in large areas.

### 5.2. Wetland Loss Assessment

The area percent of marsh and water area in Zhenlai County descended from 27.55% to 20.55% during 1985–2018 and was mainly transformed into farmland, which is consistent with other previous studies [79]. In the wetland loss assessment, occurrence frequency maps were derived to identify marsh or water areas at a certain period instead of using the raster value of each year, effectively reducing the errors caused by interpretation and avoiding the influence of misclassification in one year, as many previous works have evidenced [19,40]. There was an agglomerating distribution pattern of wetland evaluation zones, except for WDA, which was scattered in the small pools of Zhenlai County. The distribution pattern of wetland evaluation zones indicates that marsh degradation in Zhenlai County was caused by intensive farming [80], and water area degradation was caused by the water requirements of the surrounding farmland [81,82].

NDVI, mNDWI, and $\sigma^0$, which can estimate the vegetation and water conditions, were chosen to represent the ecological environment quality in different assessment zones. The ecological environment quality in different assessment zones was the combined result of the types and distribution of land use and the amount of rainfall and receding water from the paddy field. Using the frequency of wetland occurrence at a certain stage in wetland evaluation, the degradation area in this study mainly includes changes in land-use types and partly includes environmental deterioration. Therefore, the low value in mNDWI and high value in NDVI and $\sigma^0$ in the degradation area were caused by the transformation from wetland to farmland, the changes of vegetation characteristics or vegetation types [83], and the dryness trend in wetland [84,85]. Compared to the stable area, the higher value in mNDWI and lower value in NDVI and $\sigma^0$ in MRA show that the hydrological condition has recovered, yet the plant community has not fully restored to its original level [86,87]. The lower value in mNDWI and higher value in NDVI and $\sigma^0$ in MVA indicate that the hydrological condition has deteriorated, and the vegetation community has evolved from hydrophyte to xerophyte [87]. The wetland change trends vary in the trajectories of historic land use, some will be better restored, and some may continue to degrade.

*5.3. Influences of Historic Wetland Change Characteristics on Its Current States*

Based on accurate land-use maps, the wetland change trajectory and its intrinsic association are well explored. The historical land-use change trajectory currently has a different impact on the spatial distribution of wetlands. More deep-water area in WSA, more shallow-water area in WRA, and more saline-alkali land and farmland in WDA, suggest the hydrological condition was deteriorating in this order [88]. The dramatic exchange between deep-water area and shallow-water area in WVA indicates that the water area was unstable and susceptible to the surrounding environment. The land-use types in the four assessment zones of marsh were more complicated than the assessment zones of water area, given that the marsh was easily damaged or destroyed by human reclamation and climate change [89]. The small amount of marsh in the MRA and MDA zones suggests that a large area of marsh had been developing and that a small area had been restoring, which is coincident with other studies in this area [90]. Scientific wetland restoration planning is urgently needed in Zhenlai County. The temporal change characteristics of wetlands from April to October were different in different assessment zones. In the water area assessment zones, wetland changes mainly occurred in marsh and shallow-water areas. The more marsh area present before July, the more shallow-water area after July, suggesting the transition from marsh to shallow-water area; this transition coincides with rainfall and receding water from the paddy field [91], and is also appliable in MSA, MRA, and MVA, where marsh area steadily decreased from April to October and shallow-water area increased from July. With the water input to the system, the trends of marsh to shallow-water area and shallow-water area to deep-water area were clearly observed in MSA, but not obvious in degradation areas (WDA and MDA), owing to the natural situation of wetland in MSA and the manual regulation by water conservancy facilities in degradation area [92].

## 6. Conclusions

This study mainly focused on high spatial and long temporal resolution land-use mapping and explored the effect of historical land-use change trajectories on wetland variations. Fast and accurate land-use mapping is promising in GEE, even with large areas. The complementary optical and SAR indices and multitemporal analysis have significantly improved the classification accuracy of land-use mapping in 12-day intervals. The RFC classifier performed well in all land-use maps. The contrary value between the kappa index and the RFC importance information of optical and SAR data suggests the different roles of indices in RFC. From 1985 to 2018, 27% wetlands have degraded in different transfer forms. The stable, degradation, and restoration periods were identified according to the annual changes of wetlands, and the degraded, stable, restored, and vulnerable zones were assessed based on the transformation characteristics among wetlands and other land-use types. The impact of historical land-use trajectories on wetland change characteristics is currently different in land-use types and distributions, and the ecological environment quality is the comprehensive result of the effect of historical land-use trajectories and the amount of rainfall and receding water from paddy fields. Wetland loss was mainly caused by reclamation, and wetland ecological environment quality was more complicated to compare among different wetland loss assessment zones and time series as the existence of farmland. Wetland restoration, and stopping the unreasonable reclamation, are the key to wetland protection.

**Supplementary Materials:** The following are available online at https://www.mdpi.com/article/10.3390/rs13224514/s1, Table S1: The parameters derived from the Sentinel sensor, Table S2: Number of the ROI and classification accuracies during 1985 to 2018, Figure S1: Characteristics of different land-use types from 1985 to 2018, Figure S2: The land-use maps based on different data sources or methods.

**Author Contributions:** Conceptualization, S.S. and Y.C.; Funding acquisition, Y.L. and Y.H.; Investigation, S.S., Y.C., Y.L., M.L., Z.X. and B.L.; Methodology, S.S., M.L., J.M., Z.X., D.W. and T.Z.; Project administration, Y.H.; Software, S.S., Z.X., B.L. and T.Z.; Supervision, Y.H.; Validation, S.S.; Writing—original draft, S.S.; and Writing—review and editing, S.S. and Y.C. All authors have read and agreed to the published version of the manuscript.

**Funding:** This research was funded by the National Key Research and Development Project of China [2016YFC0500401].

**Data Availability Statement:** The data presented in this study are available on request from the corresponding author.

**Conflicts of Interest:** The authors declare no conflict of interest.

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
