# Peer review of "Using Time Series Optical and SAR Data to Assess the Impact of Historical Wetland Change on Current Wetland in Zhenlai County, Jilin Province, China"

_remotesensing, doi:10.3390/rs13224514_

Round 1

Reviewer 1 Report

This paper about High spatiotemporal resolution assessment for the impact of 
historical wetland loss on current wetland change trajectory using time series optical remote sensing and SAR data – a case study in Zhenlai County, Jilin Province, China falls into the category of scientific articles that try to use modern interpretation techniques to analyze the modification on wetland areas. The structure of the paper is in good connection with the objectives proposed, but before accepting for publishing the authors must try to made some modification

  1. The title of the paper is to long and requires an shortage
  2. Also the abstract is quite long and need an revising of the language used and the succession of the ideas: row 31 to row 32 must be at the introduction of the abstract. Also in the abstract the authors must mention that this site is included in the RAMSAR list.
  3. in the keyword part we think that Google Earth Engine is not necessary
  4. at the end of introduction part the authors must clearly specify the objectives of the paper.
  5. the text from row 133 to row 136 is not necessary. The description of the vegetation can be found in other specific papers but not mentioned in the references list by the authors.
  6. in the description of data used and platforms is necessary to have some references about the images used.
  7. the authors declare in rows 253-254 that To better explore the dynamics of marsh and water areas, water areas were divided into shallow-water area and deep-water area. What was the depth criterion used to divide in shallow and deep water?
  8. the description of building coherence coefficient (CC) steps (from row 290 to row 298) needs some references or is original concept of the authors
  9. the results and discussion part are connected with method used and the results obtained in similar analysis. Maybe authors will improve the reference list connected with the discussion part with papers outside China or South-East Asia. 
  10. the conclusion part must be rewritten to connect the results obtained with the effects on land used in the area and solution proposed for reducing the impact of wetland areas lost.

Reviewer 2 Report

TITLE

It is long and not clear. It is not easy to understand the meaning of the impact of historical wetlands on current wetland change trajectory(?). I should erase these last two words: you will explain into the text.

In these two first line you can use wetlands only one time: synthetize.

Before use a coma and then Synthetic Aperture Radar instead than SAR

If I’ve understood well title could sound:

High spatiotemporal resolution assessment for the impact of historical wetland loss on current one, using time series data from both optical and microwave systems – a case study in Zhenlai County, Jilin Province, China.

But I am sure that could be improved by a mother tongue.

ABSTRACT

You use Synthetic Aperture Radar on both title and key words and any into the abstract?

28        random forest classification (RFC) capital letters

31-32  The largest wetland area in Zhenlai County was 1,780.52 km2 in 1998. From 1985 (1,474.08 31 km2) to 2018 (1,078.28 km2), 40.78% of degraded wetlands were reclaimed into farmland.

Why do you give the dimension of wetland in 1998 if you cite data from 1985. It is not easy to understand the % of what?

39        remove maps

39        and addresses the relationship between characteristics of historic wetland change and its status quo (you mean the present-day situation?!)

KEYWORDS

Capital letters

68        optical remote sensing Capital letters? Doesn’t exist an acronym?

70        synthetic aperture radar Capital letters

Instead of Fig it is better Fig.

2.2.1 Remote sensing data (WITHIN THIS CHAPTER MAYBE YOU CAN TRY TO CITE THE FOLLOWING PAPER TO SHOW DIFFERENT USE OF SATELLITE IMMAGES IN DIFFERENT AREAS)

Randazzo, M. Cascio, M. Fontana, F. Gregorio, S. Lanza, A. Muzirafuti (2021) Mapping of Sicilian Pocket Beaches Land Use/Land Cover with Sentinel-2 Imagery: A Case Study of Messina Province. Land, 10, 678, 2-20. https://doi.org/10.3390/land10070678

Muzirafuti, G. Barreca, A. Crupi, G. Faina, D. Paltrinieri, S. Lanza and G. Randazzo (2020). The Contribution of Multispectral Satellite Image to Shallow Water Bathymetry Mapping on the Coast of Misano Adriatico, Italy. Journal Marine Science and Engineering, 8, 126, pp 1- 21.

208      then instead of next

211      random forest classification capitol letters

340       why to cite fig 7 now

383       I shouldn’t start with fig. 5

Fig 7 and 8 are not cited in the text

A paragraph (4.2.2 and 4.3.1) cannot start with a figure

501      why the dot after fig 11? The right way is (Fig. 11) and eventually (Fig. 11).

520-533 it is not real clear

Maybe the whole RESULTS chapter should be clarified better, using better figures and tables

Conclusion

You could mention the period 1985-2018
